REGISTERED REPORT PROTOCOL

# Determining the effects of nanoparticulate air pollution on proteostasis in *Caenorhabditis elegans*

**Emily H. Green, Elise A. Kikis**⬤*

Biology Department, The University of the South, Sewanee, TN, United States of America

* eakikis@sewanee.edu

This is a Registered Report and may have an associated publication; please check the article page on the journal site for any related articles.

## Abstract

The proteostasis network comprises the biochemical pathways that together maintain and regulate proper protein synthesis, transport, folding, and degradation. Many neurodegenerative diseases are characterized by a failure of the proteostasis network to sustain the health of the proteome, resulting in protein misfolding, aggregation, and, often, neurotoxicity. Although important advances have been made in recent years to identify genetic risk factors for neurodegenerative diseases, we still know relatively little about environmental risk factors such as air pollution. Exposure to nano-sized particulate air pollution, referred to herein as nanoparticulate matter (nPM), has been shown to trigger the accumulation of misfolded and oligomerized amyloid beta in mice. This suggests that the ability to maintain proteostasis is likely compromised in Alzheimer 's disease (AD) pathogenesis upon exposure to nPM. We aim to determine whether this aspect of the environment interacts with proteostasis network machinery to trigger protein misfolding. This could at least partially explain how air pollution exacerbates the symptoms of neurodegenerative diseases of aging, such as AD. We hypothesize that nPM challenges the buffering capacity of the proteostasis network by reducing the efficiency of folding for metastable proteins, thereby disrupting what has proven to be a very delicate proteostasis balance. We will test this hypothesis using *C. elegans* as our model system. Specifically, we will determine the impact of particulate air pollution on the aggregation and toxicity of disease-associated reporters of proteostasis and on transcriptional responses to stress.

## Introduction

The term 'proteostasis' refers to the ability of cells and organisms to maintain a healthy proteome via the activity of the many pathways and processes that comprise the proteostasis network. The proteostasis network consists of nearly 2000 proteins [1] and coordinates the regulation of protein synthesis, protein folding, protein trafficking, and protein turnover [2]. In aging, disease, and under conditions of proteotoxic stress, the proteostasis network becomes overwhelmed, resulting in proteostasis collapse and an increased load of misfolded proteins [2]. This proteostasis collapse has been documented both in young *C. elegans* engineered to express misfolded Huntington's disease (HD)-associated protein [3] and during aging [4].

**Data Availability Statement:** All relevant data from this study will be made available upon study completion.

**Funding:** Funding for this work includes Faculty Development Grants provided by the University of the South to EAK and Appalachian Colleges Association Faculty Fellowship #19 60002 to EAK. The funders have not and will not have a role in study design, data collection and analysis, decision to publish, or preparation of the manuscript.

**Competing interests:** The authors have declared that no competing interests exist.

HD is a progressive autosomal dominant neurodegenerative disorder for which the genetic determinant is an expansion of a polyglutamine (polyQ)-encoding CAG repeat in the gene that encodes the huntingtin protein [5]. PolyQ-expanded huntingtin protein misfolds and aggregates in aging individuals, and is neurotoxic.

As in HD pathogenesis, protein aggregation is also a hallmark of Alzheimer's disease (AD). For example, the amyloid precursor protein, APP, is misprocessed into a misfolded and aggregation-prone neurotoxic peptide, referred to herein as amyloid beta (Aβ). The misfolding/aggregation of Aβ precedes tau hyperphosphorylation, the formation of neurofibrillary tangles, and the onset of cognitive decline [6]. Genome-wide association studies have uncovered 29 genetic risk factors for sporadic AD [7]. ApoE4, identified in 1993, is the strongest such risk factor, associated with a two- to ten-fold increased risk for the disease compared to the general population. It likely acts via an interaction with Aβ [8,9], whose misfolding and deposition within specific regions of the brain may partially contribute to neurodegeneration [10].

Environmental risk factors for AD are also under intense investigation, pointing to the need to consider the role that gene—environment interactions play in AD progression. Underscoring the disease-relevance of gene—environment interactions, a recent epidemiological study revealed that particulate air pollution significantly exacerbates the effects of ApoE4 in women [11]. Likewise, in a mouse model of familial AD, the presence of the human ApoE4 allele was associated with increased susceptibility to nano-sized particulate matter (nPM) obtained from traffic-derived air pollution, leading to an increase in Aβ aggregation [11]. These important findings suggest that the ability to maintain proteostasis is likely compromised upon exposure to particulate air pollution.

Consistent with this hypothesis, it was recently shown that the expression levels of proteostasis network genes in *C. elegans* are responsive to nPM [12]. Furthermore, the degradative pathways of the proteostasis network have also been shown to be activated in mice exposed to nPM [13]. The effects of this dysregulation on the folding, or misfolding, of disease-associated proteins has never been directly tested. We therefore propose testing the hypothesis that exposure to nPM will challenge the buffering capacity of the proteostasis network, thereby reducing the efficiency of disease-associated protein folding. We will use wild type *C. elegans* and strains that have been engineered to express aggregation-prone disease-associated proteins to determine:

1. Whether exposure to nPM exacerbates the formation of large visible amyloid beta (Aβ) or polyglutamine (polyQ) protein aggregates in the genetically tractable *C. elegans*.

2. Whether chronic exposure to nPM triggers an increase in polyQ or Aβ toxic oligomers in *C. elegans*.

3. Whether alterations in chaperone gene expression in response to chronic nPM stress can explain the observed effects on proteostasis in *C. elegans*.

It is important to note that neuroinflammation is induced upon nPM exposure [14], making it difficult to ascertain whether effects of nPM on proteostasis are upstream or downstream of inflammation. Utilizing *C. elegans* as our model system will allow us to separate these two processes because this model nematode lacks the transcription factor NFκB and thus does not experience a canonical inflammatory response. Therefore, we will be able to examine the impact of particulate air pollution on proteostasis without the confounding effects of inflammation. Furthermore, the tools and sensors required to monitor changes in *C. elegans* proteostasis in real-time are plentiful and well-documented, making this animal an especially powerful tool for the proposed study.

## Methods

### *C. elegans* strains, growth, and maintenance

The following *C. elegans* strains will be utilized: N2 (wild type, Bristol), AM140 (rmIs132 [*unc-54*::polyQ35::YFP]) [15], AM141 (rmIs133 [*unc-54*::polyQ40::YFP]) [15], OG412 (drIs20 [vha-6p::Q44::YFP]) [16] and GMC101 (dvIs100 [*unc54*p::Abeta1-42]) [17]. All strains of *C. elegans* will be obtained from the *Caenorhabditis* Genetics Center (University of Minnesota) and maintained at 20˚C on Nematode Growth Media (NGM) seeded with OP50 *E. coli* bacteria as a food source according to the standard methods [18].

### Acquiring nanoparticulate matter from polluted air

Traffic-derived nanoparticulate air pollution samples (nPM) were collected in Los Angeles, California using a high-volume ultrafine particulate (HVUP) sampler with a Teflon filter. Dried nPM samples were eluted to 150 µg/mL with deionized water according to established methods [19]. Characterized and validated nPM samples have been generously gifted to us by the laboratory of Caleb Finch of the University of Southern California. As bacterial contamination could affect proteostasis, nPM samples will be sterilized by UV-C irradiation in a biosafety cabinet for at least 15min. To ensure that samples are free of bacterial contamination, 20uL of nPM will be transferred to an NGM plate, incubated for 3 days and examined for the appearance of colonies. For consistency, a single batch of eluted nPM will be used for all of the experiments proposed herein.

### Exposure paradigm

*C. elegans* will be grown to the L1 or L4 stage, at which point 20–50 animals will be exposed to 75ug/mL nPM. The exact number of animals exposed will depend on the constraints of specific experiments as described in our experimental design. M9 will serve as our negative control, while 5mM paraquat (PQ) will be a positive control for oxidative stress. Exposures will be performed at 20˚C in 96 well plates. 100µL of 2X M9 liquid medium will be diluted with an equal volume of nPM (or water for unexposed controls) supplemented with 10µg/mL cholesterol and OP50 bacteria for up to 3 days. To maintain uniform nPM concentration for the duration of the experiment, and to prevent animals from falling to the bottom of the well, 96-well plates will undergo continuous gentle rocking on a nutator. All exposures will be performed in biological triplicate.

### Counting aggregates

For YFP-tagged polyQ-expressing animals, at least 20 individuals for each exposure will be singled onto seeded plates and chilled on ice to slow movement. Aggregates will be counted in individual live animals using a fluorescent stereomicroscope (Leica M165 FC) fitted with a digital camera according to established methods [20]. All experiments involving aggregate counting will be performed in at least biological triplicate.

Amyloid beta (Aβ) is not fluorescently tagged; therefore, nPM-mediated changes in aggregation propensity will be determined via immunofluorescence as previously described [3]. In short, animals will be fixed with paraformaldehyde, treated with β-mercaptoethanol and collagenase, and incubated with the monoclonal anti-amyloid beta antibody derived from clone BAM-10 sold by Sigma (St. Louis, MO). Secondary antibodies will be labeled with FITC for visualization on a Leica sp7 laser scanning confocal microscope, or with a fluorescent compound microscope (Zeiss Axio observer).

## Thrashing assays

To assay for changes in proteotoxicity triggered by exposure to nPM, thrashing assays will be performed as described previously [21]. Specifically, 30 L4 AM140 or AM141 animals expressing polyQ35-YFP or polyQ40-YFP in body wall muscle cells will be exposed to nPM in liquid for 72hrs at which time they will be allowed to recover on seeded plates for 15-30min and then picked onto a drop of M9. After a 30s recovery, the number of body bends per minute will be counted manually under a Leica M165 FC stereomicroscope. All thrashing assays will be performed in at least biological triplicate.

## Paralysis assays

To assay for changes in Aβ toxicity in body wall muscle cells in response to nPM, GMC101 animals will be exposed to +/- nPM either as L4s for 1hr (acute stress) or as L1s for 3d (chronic stress). Paralysis will then be monitored at 20˚ or 25˚C for at least 3 days. N2 animals will be utilized as a control to ascertain whether any observed paralysis is due to a gene (Aβ)—environment (nPM) interaction or is simply an effect of the nPM, irrespective of Aβ. All paralysis assays will be performed in at least biological triplicate.

## qRT-PCR

RNA will be isolated using 250μL Trizol (Sigma-Aldrich, St. Louis, MO) according to the manufacturer's instructions. Removal of genomic DNA and cDNA synthesis will be performed with the iScript gDNA clear cDNA synthesis kit (Bio-Rad, Hercules, CA). qPCR will be performed with the SYBR green master mix (Bio-Rad, Hercules, CA) using previously published gene-specific primers (**Table 1**). To control for differences in sample concentration, the expression of stress genes will be normalized to actin. We will plot gene expression relative to the M9 control and perform t-tests to compare each exposure to the control. All gene expression studies will be performed in at least biological triplicate.

## Native gels

Native protein from ~50 animals will be extracted mechanically by grinding in liquid nitrogen followed by the addition of 30μL of ice cold native lysis buffer as described previously [24]. Native samples will be resolved immediately after extraction on a 6% native PAGE gel. Fluorescent bands containing YFP protein will be detected under UV light on a BioRad gel-doc imaging system (Hercules, CA). YFP-containing polyQ protein bands representing either monomeric protein or high molecular weight species will be quantified using ImageJ and the ratio of monomers to high molecular weight species will be calculated. Detection of unlabeled Aβ$_{1-42}$ oligomers will be as described above for polyQ, but with some modifications. Specifically, after electrophoresis, gels will be heated in SDS to denature the resolved protein and then

**Table 1. Primers used to investigate changes in gene expression.**

| Gene | Forward Primer | Reverse Primer | References |
|---|---|---|---|
| hsp-4 | CTAAGATCGAGATCGAGTCACTC | GCTTCAATGTAGCACGGAAC | Haghani et. al., 2019 [12] |
| gst-4 | GATGCTCGTGCTCTTGCTG | CCGAATTGTTCTCCATCGAC | Haghani et. al., 2019 [12] |
| hsp-6 | TCGTGAACGTTTCAGCCAGA | CTCAGCGGCATTCTTTTCGG | Bennet et. al., 2014 [22] |
| C12C8.1 | ACGGGCTTTCCTTGTTTT | ACTCATGTGTCGGTATTTATC | Prahlad et. al., 2008 [23] |
| F44E5.4 | TGTCCTTTCCGGTCTTCCTTTTG | AATGAACCAACTGCTGCTGCTCTT | Prahlad et. al., 2008 [23] |
| Actin | ATCACCGCTCTTGCCCCATC | GGCCGGACTCGTCGTATTCTT | Prahlad et. al., 2008 [23] |

a western transfer to a PVDF membrane will be performed as described [24]. Standard immu-noblot protocols with anti-Aβ antibodies (clone BAM-10, Sigma) will allow visualization using the LI-COR Odyssey system (Lincoln, NE). All native gels will be performed in at least biologi-cal triplicate.

**SDS-PAGE.**   Total protein will be isolated from 10–20 individuals by boiling in laemmli sample buffer. Samples will be run on a 10% SDS-PAGE gel and transferred to a PVDF mem-brane. Immunodetection will be performed with anti- anti-Aβ antibody (clone BAM-10, Sigma), anti-YFP antibody (Rockland), anti-polyQ antibody (clone 3B5H10, Sigma), or anti-alpha-tubulin antibody (B-5-1-2, Sigma) as a loading control. All secondary antibodies will be IR-conjugated for visualization using the LI-COR Odyssey system (Lincoln, NE). All SDS-PAGE gels will be performed in at least biological triplicate.

## Experimental design and rationale

### A. Does the exposure to nPM exacerbate the formation of large visible amyloid beta (Aβ) or polyglutamine (polyQ) protein aggregates in the genetically tractable *C. elegans*?

**Background/Rationale.**   AD-associated fragments of Aβ have been expressed in *C. elegans* and shown to display age-dependent and temperature-dependent toxicity [17]. Likewise, polyQ-containing proteins have been expressed in *C. elegans* muscle cells [15], intestines [16] or neurons [25], where they aggregate and are toxic in a manner dependent both on age and the length of the polyQ repeat expansion. The polyQ-expressing animals in particular have been extraordinarily sensitive reporters of the protein folding environment, as various stresses such as increased misfolded protein load [3] or aging [15] trigger premature polyQ aggrega-tion. If nPM causes generalized damage to cellular proteins that is of sufficient magnitude to stress the proteostasis network, then we would expect to observe an increase in Aβ or polyQ aggregation.

**Experiment.**   To investigate the effects of exposure to nPM on disease-associated protein folding in the intestine, the initial site of contact with nPM, we will expose 20–30 OG412 (Q44-YFP) animals at the L4 larval stage to nPM for three days and the formation of large visi-ble protein aggregates will be monitored as described above. These exposure assays will be per-formed in at least biological triplicate. Results from each replicate will be pooled and the number of large visible aggregates in the intestines of individual animals will be plotted along with mean and standard error of the mean (SEM).

To determine whether nPM exposure affects proteostasis in more distal tissues, such as the body wall muscle cells, AM140 (Q35-YFP), AM141 (Q40-YFP), or GMC101 (Aβ$_{1-42}$) animals will be exposed to nPM at the L1 or L4 stage and the formation of large visible protein aggre-gates will be monitored as described above. Exposures will be performed in at least biological triplicate. Results from each replicate will be pooled and the number of large visible aggregates in body wall muscle cells of individual animals will be plotted along with mean and SEM.

As an additional assay for changes in the relative concentration of large visible aggregates in response to nPM, SDS-PAGE gels will be performed as described above. This is expected to reveal the formation of SDS-insoluble protein species.

### B. Does exposure to particulate air pollution trigger an increase in polyQ or Aβ toxic oligomers in *C. elegans*?

**Background/Rationale.**   While large visible deposits of aggregated protein are hallmarks of many neurodegenerative diseases, overwhelming evidence points to these large aggregates

being cytoprotective, with small oligomers being the toxic species [26]. Thus, while the large aggregates examined in the previous aim may be indicative of proteostasis imbalance, they cannot be taken as evidence of proteotoxicity. Therefore, we will determine whether exposure of *C. elegans* to nPM alters the relative amounts of soluble, aggregated, and oligomeric protein and whether this leads to proteotoxicity.

**Experiment.**   To determine whether nPM or PQ exposure alters the aggregation profile in animals expressing expanded polyQ or Aβ, 50 AM140 (muscle Q35-YFP), AM141 (muscle Q40-YFP), GMC101 (muscle Aβ$_{1-42}$) or OG412 (intestinal Q44-YFP) animals will be exposed as L4s to nPM or PQ as described above. High molecular weight species, likely representing oligomers, will be quantified via native gel electrophoresis and compared to the abundance of monomers. Exposures and native gels will be performed in triplicate and mean ratios will be calculated and plotted. T-tests will be performed with GraphPad Prism comparing the ratios observed in control (M9) samples to those observed following exposure to nPM or PQ.

Toxicity of polyQ35, polyQ40, or Aβ in body wall muscle cells will be measured as a function of thrashing rate in liquid as described previously [21]. Because toxicity is only observed for the Aβ-expressing strain (GMC101) under elevated temperature [17], it will be interesting to determine whether nPM, like thermal stress, causes sufficient proteostatic stress to expose Aβ-toxicity. Exposures and thrashing assays will be performed in biological triplicate. Results from each replicate will be pooled and data from individual animals will be plotted along with mean and SEM.

## C. Can alterations in chaperone gene expression in response to chronic nPM stress explain the observed effects on proteostasis in *C. elegans*?

**Background/Rationale.**   As cells and organisms are exposed to conditions that trigger protein misfolding, transcriptional responses involved in protein quality control, such as the heat shock response (HSR), the Unfolded Protein Response (UPR), and the oxidative stress response, are induced [27–29]. If particulate air pollution is a significant source of proteotoxic stress, we would expect it to trigger stress-responsive gene expression. In fact, the immediate transcriptional upregulation of some UPR targets such as the molecular chaperone gene *hsp-4* and oxidative stress targets such as the antioxidant gene, *gst-4* [12] has been recently reported for L1-stage wild type *C. elegans* exposed to nPM for 1hr. As at least some of the studies proposed here will involve chronic (3d) exposures to nPM, we aim to determine whether the expression of proteostasis network components responds to nPM stress over this time period. Such gene expression changes would be expected to underlie any observed changes in the protein folding environment revealed via the completion of this study.

**Experiment.**   As exposure to particulate air pollution is presumably a chronic event that leads to protein damage over time, we will examine the transcriptional effects of chronic (72hr) exposure to nPM. Specifically, we will expose 50 L4 stage wild type (N2) animals to nPM or PQ as described above. M9 will serve as the negative control (no stress). PQ will serve as a positive control for oxidative stress, as nPM has been shown to trigger this stress response [12,30]. Samples will be harvested after 1hr, 24hrs, or 72hrs of exposure for RNA isolation. qRT-PCR will be utilized to monitor the expression of the HSR targets, C12C8.1 and F44E5.4, the UPR target *hsp-4*, the mitochondrial UPR target hsp-6, and the oxidative stress response target *gst-4*. Gene expression changes will be represented relative to the basal levels of gene expression (M9 sample) at each time point. Exposures will be performed a minimum of three times, resulting in at least biological triplicates for each time point. All qRT-PCR reactions will be prepared in technical triplicate to control for pipetting error.

## Timeline

It is our estimation that the proposed gene expression analysis and the examination of the effects of nPM exposure on polyQ-expressing animals will be completed by the middle of 2021. Analysis of Aβ-expressing animals will commence near the middle of 2021 and will take approximately one year. Therefore, we expect to have this study completed and a manuscript ready for submission by the end of 2022.

## Dissemination of results

Following the completion of the study proposed here, the results will be published as a Registered Report in PLoS ONE.

## Author Contributions

**Conceptualization:** Elise A. Kikis.

**Funding acquisition:** Elise A. Kikis.

**Methodology:** Elise A. Kikis.

**Project administration:** Elise A. Kikis.

**Supervision:** Elise A. Kikis.

**Writing – original draft:** Emily H. Green.

**Writing – review & editing:** Elise A. Kikis.

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
