## [Decision Letter · Decision Letter 0]

25 Sep 2020

PONE-D-20-23642

Determining the effects of nanoparticulate air pollution on proteostasis in Caenorhabditis elegans

PLOS ONE

Dear Dr. Kikis,

Thank you for submitting your manuscript to PLOS ONE. I apologize for the delay in obtaining a review, but I believe that you now have some useful comments to consider. After careful consideration, we feel that it has merit but does not fully meet PLOS ONE’s publication criteria as it currently stands. Therefore, we invite you to submit a revised version of the manuscript that addresses the points raised during the review process.

The main questions to address are related to your choice of methodology that you will find below.  There were several other minor issues to address.

We look forward to receiving your revised manuscript.

Kind regards,

David R Borchelt

Academic Editor

PLOS ONE

Journal Requirements:

Reviewers' comments:

Reviewer's Responses to Questions

**Comments to the Author**

1. Does the manuscript provide a valid rationale for the proposed study, with clearly identified and justified research questions?

Reviewer #1: Yes

Reviewer #2: Partly

2. Is the protocol technically sound and planned in a manner that will lead to a meaningful outcome and allow testing the stated hypotheses?

Reviewer #1: Yes

Reviewer #2: Partly

3. Is the methodology feasible and described in sufficient detail to allow the work to be replicable?

Reviewer #1: Yes

Reviewer #2: Yes

4. Have the authors described where all data underlying the findings will be made available when the study is complete?

Reviewer #1: Yes

Reviewer #2: Yes

5. Is the manuscript presented in an intelligible fashion and written in standard English?

Reviewer #1: Yes

Reviewer #2: Yes

6. Review Comments to the Author

You may also provide optional suggestions and comments to authors that they might find helpful in planning their study.

Reviewer #1: This is an exciting study. Please address minor concerns:

The Registered Report Study by Green and Kikis titled “Determining the effects of nanoparticulate air pollution on proteostasis in Caenorhabditis elegans” explores the impact of air pollution on organismal proteostasis. Nano-sized particulate matter (nPM) is known to contribute to the pathogenesis of neurodegenerative diseases across mouse models and in humans. Moreover, previous research using mice and C. elegans suggests that nPM affects host proteostasis. The authors pose to investigate the effect of nPM on the host by posing an interesting hypothesis “that exposure to nPM will challenge the buffering capacity of the proteostasis network, thereby reducing the efficiency of disease-associated protein folding.” C. elegans is an ideal model organism to test this hypothesis as it offers a comprehensive collection of neurodegenerative disease models, proteostasis reporters, and genetic tools. Moreover, the model lacks inflammatory response and can therefore be used to study a direct effect of nPM on proteostasis. Overall, the authors propose experiments that will explore a novel area of research that has been recently gaining interest within the scientific community. If successful, the results of this study can be extrapolated to and followed up in higher organisms. This study will provide exciting results and is recommended for publication in PLoS One. There are minor comments that need to be addressed:

• Why is nano-sized particulate air pollution abbreviated as nPM? What does M stand for? Missing “matter”?

• The threshold of polyQ aggregation is between 35-40. The authors may benefit by including polyQ37 strain instead of, or in addition to, polyQ40.

• It is assumed that the animals will ingest nPM, and the immediate effect will be on the proteostasis in the intestine. The study may benefit from including intestinal polyQ44. If the particulates are restricted to the intestine, muscle-specific polyQs may not be affected; however, that would not mean there is no effect.

• Are the nPM samples sterile? If not, will they be sterilized by UV, heat, or any other method? If not sterile, the bacteria in the sample may affect proteostasis.

• Line 108: change the citation style to numerical so it matches the rest of the paper.

• Line 108: with deionized water?

• If the stock nPM concentration is 150ug/mL and the animals will be exposed to 75ug/mL that means the stock will have to be diluted 1:1 to obtain the final concentration. The protocol states that 200uL of M9 supplemented with 10ug/mL cholesterol will be present in each well. It may be more clear to write that 100uL of 2X M9 at 20ug/mL cholesterol will be diluted with an equal volume of nPM.

• For aggregate counting, will only one biological replicate be performed?

• For qRT-PCR, it would be good if the authors could include mitochondrial hsp70 (hsp-6) as well. That way, all major stress pathways are assessed.

• Line 174: please add particulates “…exposure to air pollution particulates.”

• The authors have to be cautious about interpreting aggregate counts when normalized to body size. Although the animals may be smaller in size, they are not developmentally delayed (post L4) and their biological age does not change. The absolute numbers of aggregates may be a better approximation of the actual effect of nPM.

• Line 198: Please write out standard error of the mean and then abbreviate.

• The timeline seems appropriate

Reviewer #2: This Registered Report Protocol by Green and Kikis outlines an interesting set of experiments aimed at understanding the effects of nano-sized particulate air pollution (nPM) on proteostasis using C. elegans. While the general premise of the study will be of interest to a wide audience, the motivations for this particular protocol require clarification, the methods in some cases require modification, and the overall organization of the study can be improved.

Major comments:

Overall, the organization of the study goals appears to be in reverse order. Establishing nPM-induced phenotypic changes, i.e. motor dysfunction and protein aggregation, should precede the mechanistic investigation of chaperone gene changes that may explain these phenotypes. The authors should either re-organize the study, or provide clear and explicit justification for the order of experiments chosen.

Aggregates simply cannot be counted by eye on a stereo-microscope. This is neither accurate nor reproducible. Individual protein aggregates are only grossly visible using the method described, and are indistinguishable along the 3-dimensional axis of the animal. The method for fluorescent aggregate quantification as described is simply unacceptable.

A single thrashing assay (manual counting of body bends) is proposed for measurement of motor defects. However, manual quantification is prone to error, and using only a single assay is unlikely to produce meaningful results. Instead, the authors should consider video capture of thrashing worms followed by automated analysis of multiple motor parameters as a much more powerful assessment of motor function.

The authors propose to use Native gels for biochemical analysis of aggregates. SDS-PAGE should be performed as well, in order to resolve SDS- and heat-stable aggregate species.

Minor comments:

In the introduction, the authors should be careful not to give the impression that the mechanisms of HD or AD are entirely known. The statement that AD is more mechanistically complex than HD is not warranted, given that both diseases involve protein aggregation and the relationship of aggregates with neurotoxicity is still poorly understood in both cases. The authors should introduce the diseases and state what is as yet unknown, in order to set up the motivation for their study.

The authors should not overstate the evidence for the link between air pollution and AD onset as being “well-established” (line 69). The authors should simply cite the relevant literature, describing specifically what has been found that supports this link.

It is not clear from the introduction precisely what is still not known regarding the effects of nPM on proteostasis. For example, the authors cite a C. elegans study in which nPM induced changes in proteostasis network genes, yet the first goal outlined for the current study is to determine “Whether chaperone gene expression is altered in C. elegans exposed to chronic nPM stress.” Aren’t the changes in chaperone gene expression in response to nPM already known from the cited reference? Please clarify how the proposed goal is novel.

The second goal outlined in the introduction reads “Whether the folding of amyloid beta (Aβ) or polyglutamine (polyQ) proteins” but the authors intend to say “Whether the misfolding…” Please correct.

It is not clear how the second goal (whether the misfolding of disease-linked proteins is exacerbated with nPM) and the third goal (whether nPM induces toxic oligomers) are different. Is the second goal looking at inclusions rather than oligomers? Is the distinction between the goals related to toxicity? Please revise such that the two goals are clearly delineated.

In line 108, I believe “deionized” should read “deionized water”. Please correct.

The authors should state whether negative control wells will have equal volume of the same solvent that is used to make up the working solution of nPM.

The authors should consider using complete S medium instead of M9 for liquid culture of C. elegans, since S medium contains additional nutrients to support C. elegans survival.

7. PLOS authors have the option to publish the peer review history of their article (what does this mean?). If published, this will include your full peer review and any attached files.

Reviewer #1: **Yes: **Daniel Czyz

Reviewer #2: No

---

## [Author Response · Author response to Decision Letter 0]

6 Nov 2020

1. Reviewer #1 pointed out that we did not define the letter “M” in “nPM.” This has been corrected. Specifically, we have included the following sentence in the introduction “…nano-sized particulate matter (nPM) obtained from traffic-derived air pollution…”

2. Reviewer #1 suggested that we “may benefit by including polyQ37 instead of, or in addition to, polyQ40.” This was an interesting suggestion and one that we seriously considered. In the end, we would like to stick to polyQ35 and polyQ40. This is because the polyQ40 strain has already been shown to be highly sensitive to changes in the protein folding environment. Namely, Gidalevitz et. al. demonstrated this is in Science 311 (5766), 1471-1474. More recently, this same protein was shown to be sensitive to changes in the genetic background BMC biology 18 (1), 1-20. Taken together, we think that polyQ40 is a very important strain to use in our proposed project. Fewer studies have utilized polyQ37. While we could theoretically use all three strains, we feel that that would be excessive and would needlessly use up too much of our stock of nPM, especially as we are also including in our study a strain expressing polyQ44 in the intestine and a strain expressing Abeta in body wall muscle cells.

3. Reviewer #1 asked whether the nPM samples are sterile. Based on this comment, we have included the following sentence in the methods section, “As bacterial contamination could affect proteostasis, nPM samples will be sterilized by UV-C irradiation in a biosafety cabinet for at least 15min. To ensure that samples are free of bacterial contamination, 20uL of nPM will be transferred to an NGM plate, incubated for 3 days and examined for the appearance of colonies.”

4. Reviewer #1 suggested that we clarify the way our nPM will be diluted to obtain the appropriate final concentrations of nPM and Reviewer #2 also asked about the solvent for nPM and negative controls (I am paraphrasing these comments/suggestions): To address these concerns, we have added the following sentence to the methods section, “Exposures will be performed at 20°C in 96 well plates. 100μL of 2X M9 liquid medium will be diluted with an equal volume of nPM (or water for unexposed controls) supplemented with 10μg/mL cholesterol and OP50 bacteria for up to 3 days.” It should be noted that nPM is already in water, so there are no solvent considerations beyond those addressed in the sentence above.

5. Reviewer #1 asked whether only one biological replicate will be performed for aggregate counting. All experiments in this study will be performed in at least biological triplicate. We have now clarified this throughout the text (in the section on aggregate counting and other sections as well). 

6. Reviewer #1 cautioned us not to normalize aggregate counts to body size. Thank you for this suggestion. The text has been revised to remove any normalization to body size. 

7. Reviewer #2 expressed concern about the organization of the study. Specifically, this reviewer suggested that “establishing nPM-induced phenotypic changes should precede the mechanistic investigation of chaperone gene expression.” This point is well-taken and we have rearranged the aims accordingly.

8. Reviewer #2 stated that “aggregates simply cannot be counted by eye on a stereo-microscope.” To address this concern, we have now indicated that “Aggregates will be counted in individual live animals using a fluorescent stereomicroscope (Leica M165 FC) fitted with a digital camera according to established methods (20).” The polyQ aggregates are very large, bright, and defined. Stereomicroscopes, albeit fitted with nice digital cameras, have been used successfully in many studies to quantify aggregate number. These studies include the original paper in which the muscle polyQ model was studied (Morley et. al., (2002) PNAS) and a more recent report of the effects of genetic background on polyQ aggregation (Alexander-Floyd et. al. (2020) BMC Biology). The methods we are proposing here mirror those of the more recent study.

9. Reviewer #2 suggested that we “consider video capture of thrashing worms followed by automated analysis of multiple motor parameters.” We are aware that recent technological advances have made such automated analysis increasingly common. If cost were not prohibitive, we would (gladly and enthusiastically) purchase a MicroTracker or similar device. However, at $16,400, this far exceeds our budget. Fortunately, our lab and others have utilized manual counting of thrashing effectively to observe changes in motility for these and similar animals. Thus, we are cautiously optimistic that, even with our proposed methods, we will be able to observe decreases in muscle function.

10. Reviewer #2 suggested that “SDS-PAGE should be performed… in order to resolve SDS- and heat-stable aggregate species.” We have now included this in the section on the characterization of large, visible, protein aggregates. Specifically, we state “As an additional assay for changes in the relative concentration of large visible aggregates in response to nPM, SDS-PAGE gels will be performed as described above. This is expected to reveal the formation of SDS-insoluble protein species.”

11. Reviewer #2 suggested that we “be careful not to give the impression that the mechanisms of HD and AD are entirely known” and also “not overstate the evidence for the link between air pollution and AD onset being ‘well-established.’” These are good points and the introduction has been revised accordingly. 

12. Reviewer #2 suggests that “it is not clear from the introduction precisely what is still not know regarding the effects of nPM on proteostasis.” Thank you for mentioning that, because this is a very important point. In fact, very little is known about the effects of nPM on proteostasis and that is why we think that this study is so important. Specifically, there have only been two studies that touch on this at all and neither examined the effects of nPM on protein folding. To clarify this point, we have revised the introduction to now say, “Consistent with this hypothesis, it was recently shown that the expression levels of proteostasis network genes in C. elegans are responsive to nPM (12). Furthermore, the degradative pathways of the proteostasis network have also been shown to be activated in mice exposed to nPM (13). The effects of this dysregulation on the folding, or misfolding, of disease-associated proteins has never been directly tested. We therefore propose testing the hypothesis that exposure to nPM will challenge the buffering capacity of the proteostasis network, thereby reducing the efficiency of disease-associated protein folding.”

13. Reviewer #2 pointed out that “it is not clear how the second goal and the third goal are different” and asked whether “the second goal is looking at inclusions rather than oligomers.” That is exactly right. What is now aim “A” is examining large visible aggregates and what is now aim “B” is examining oligomers. We have also included an analysis of toxicity in with aim B because of the likely connection between oligomers and toxicity. To clarify this, we have reworded aim A as follows, “Does the exposure to nPM exacerbate the formation of large visible amyloid beta (Aβ) or polyglutamine (polyQ) protein aggregates in the genetically tractable C. elegans?” We have likewise reworded aim B, “Does exposure to particulate air pollution trigger an increase in polyQ or Aβ toxic oligomers in C. elegans?

14. Reviewer #2 suggested that we use complete S medium instead of M9. We had originally proposed using M9 because the one prior C. elegans study involving exposure to nPM utilized M9 and we wanted to be as consistent with that study as possible so as to compare results. However, it is worth noting that the prior study did short exposures, so media would have admittedly less of an effect on C. elegans growth and development. Therefore, we did some pilot studies in our lab to ensure that our chronic exposure in M9 does not negatively affect the animal. We found that under our proposed conditions, animals reach adulthood as expected and are phenotypically normal—meaning that they lay fertilized eggs and the eggs hatch as expected. There are no obvious developmental delays, bagging, etc. Thus, we propose sticking with the M9-based media supplemented with cholesterol and OP50.

15. Both reviewers identified some sentences with missing or incorrect words. Thank you for pointing these out! We have corrected them in the revised manuscript.

---

## [Decision Letter · Decision Letter 1]

23 Nov 2020

Determining the effects of nanoparticulate air pollution on proteostasis in Caenorhabditis elegans

PONE-D-20-23642R1

Dear Dr. Kikis,

We’re pleased to inform you that your manuscript has been judged scientifically suitable for publication and will be formally accepted for publication once it meets all outstanding technical requirements.

Kind regards,

David R Borchelt

Academic Editor

PLOS ONE

Additional Editor Comments (optional):

Reviewers' comments:

Reviewer's Responses to Questions

**Comments to the Author**

1. Does the manuscript provide a valid rationale for the proposed study, with clearly identified and justified research questions?

Reviewer #2: Yes

2. Is the protocol technically sound and planned in a manner that will lead to a meaningful outcome and allow testing the stated hypotheses?

Reviewer #2: Yes

3. Is the methodology feasible and described in sufficient detail to allow the work to be replicable?

Reviewer #2: Yes

4. Have the authors described where all data underlying the findings will be made available when the study is complete?

Reviewer #2: Yes

5. Is the manuscript presented in an intelligible fashion and written in standard English?

Reviewer #2: Yes

6. Review Comments to the Author

You may also provide optional suggestions and comments to authors that they might find helpful in planning their study.

Reviewer #2: The authors have carefully revised and re-organized the manuscript, and all of my concerns have been addressed.

7. PLOS authors have the option to publish the peer review history of their article (what does this mean?). If published, this will include your full peer review and any attached files.

Reviewer #2: No

---

## [Editor Report · Acceptance letter]

26 Nov 2020

PONE-D-20-23642R1 

Determining the effects of nanoparticulate air pollution on proteostasis in *Caenorhabditis elegans*

Dear Dr. Kikis:

I'm pleased to inform you that your manuscript has been deemed suitable for publication in PLOS ONE. Congratulations! Your manuscript is now with our production department. 

Kind regards, 

on behalf of

Prof. David R Borchelt 

Academic Editor

PLOS ONE